# Neuropsychological Functions and Audiological Findings in Elderly Cochlear Implant Users: The Role of Attention in Postoperative Performance

Ilaria Giallini [1], Bianca Maria Serena Inguscio [1,2,3], Maria Nicastri [1,2], Ginevra Portanova [1,2], Andrea Ciofalo [1], Annalisa Pace [1,4], Antonio Greco [1], Hilal Dincer D'Alessandro [5] and Patrizia Mancini [1,*]

[1] Department of Sense Organs, University Sapienza of Rome, Policlinico Umberto I, Viale dell'Università 31, 00161 Rome, Italy
[2] Department of Human Neuroscience, Sapienza University of Rome, Viale dell'Università 30, 00161 Rome, Italy
[3] BrainSigns Srl, Via Tirso 14, 00198 Rome, Italy
[4] Advanced Surgical Technologies PhD Program, University Sapienza of Rome, 00185 Rome, Italy
[5] Department of Audiology, Faculty of Health Sciences, Hacettepe University, 06230 Ankara, Turkey
[*] Correspondence: p.mancini@uniroma1.it

**Abstract:** Objectives: The present study aimed to investigate in a group of elderly CI users working memory and attention, conventionally considered as predictors of better CI performance and to try to disentangle the effects of these cognitive domains on speech perception, finding potential markers of cognitive decline related to audiometric findings. Methods Thirty postlingually deafened CI users aged >60 underwent an audiological evaluation followed by a cognitive assessment of attention and verbal working memory. A correlation analysis was performed to evaluate the associations between cognitive variables while a simple regression investigated the relationships between cognitive and audiological variables. Comparative analysis was performed to compare variables on the basis of subjects' attention performance. Results: Attention was found to play a significant role in sound field and speech perception. Univariate analysis found a significant difference between poor and high attention performers, while regression analysis showed that attention significantly predicted recognition of words presented at Signal/Noise +10. Further, the high attention performers showed significantly higher scores than low attentional performers for all working memory tasks. Conclusion: Overall findings confirmed that a better cognitive performance may positively contribute to better speech perception outcomes, especially in complex listening situations. WM may play a crucial role in storage and processing of auditory-verbal stimuli and a robust attention may lead to better performance for speech perception in noise. Implementation of cognitive training in auditory rehabilitation of CI users should be investigated in order to improve cognitive and audiological performance in elderly CI users.

**Keywords:** cochlear implantation; cognition; older adults; speech perception; attention; working memory

## 1. Introduction

In 2017, the Lancet Commission of Dementia Prevention, Intervention, and Care reported a new model of dementia risk [1,2] stating that hearing impairment may account for 8% of dementia cases [3]. Indeed, hearing loss (HL) is considered the biggest controllable risk factor for dementia, potentially reducing its risk by 9% [1]. This model originates from the growing evidence over the last decade highlighting the significant associations between presbycusis, deterioration of cognitive functions, and incident dementia [4]. The risk of incident dementia has been estimated to be 2 to 5 times greater in people with mild to severe HL than in the normal hearing population [5,6].

Over the past two decades, theoretical classes of hypotheses have been developed to define the link between HL and cognitive decline. However, as pointed out by numerous

studies, e.g., refs. [7–9], we have still not clearly defined the mechanisms and direction of this link, despite remarkable progress in neuropsychology and neuroaudiology. With the aim of providing a concise theoretical framework, possible causal and non-causal mechanisms for the association between HL and cognitive decline are summarized in Table 1a,b.

**Table 1.** Causal and non-causal mechanisms linking HL and Cognitive Decline.

| **(a) Main Hypotheses for the Relationship between HL and Cognitive Decline** | |
|---|---|
| Cognitive Load Hypothesis [4] | The cognitive load hypothesis suggests that HL leads to greater sensory-perceptual effort because of the incoming degraded auditory signal. The greater cognitive resources required for auditory perceptual processing have negative effects on cognitive, attentional, and mnemonic resources. In other words, cognitive decline in hearing-impaired adults might be a consequence of an overinvestment of brain activity in auditory and spoken language processing, resulting in a significant detriment to other cognitive processes. |
| Information Degradation Hypothesis [10] | The "information degradation hypothesis" suggests that degradation of stimuli (noisy environment, decrease in auditory sensitivity) requires an additional effort: as a consequence, cognitive resources used for signal codification are not available for cognitive functions. |
| Sensory Deprivation Hypothesis/Cascade Hypothesis [6,11] | According to the "sensory-deprivation hypothesis", HL demands increased cognitive effort which results in depleting cognitive performance over time. Subsequently, cognitive performance deterioration leads to social isolation that in turn causes gradual cognitive decline. Cognitive decline is believed to be potentially remediable with rehabilitation. |
| Common Cause Hypothesis [12] | Presbycusis and cognitive impairment might be signs of a common neurodegenerative process. So, sensory functioning could be a strong late-life predictor of individual differences in intellectual functioning and could be seen as an indicator of the physiological integrity of the aging brain. |
| **(b) Possible non-causal mechanisms linking HL to cognitive decline** | |
| Testing bias [7–10] | Poor verbal communication associated with HL may confound cognitive testing. HL may influence neuropsychological testing more than cognition per se. HL may introduce a systematic bias into neuropsychological assessments that are mostly designed and validated for verbal instructions and/or the presentation of stimuli. Greater sensitivity of tests in one domain (hearing or cognition) could identify deficits in that domain prior to the other one, leading to the appearance of an illusory causal relationship. |
| Conceptual bias [7–10] | Upstream common causes with no conditions causally related to others. HL brings older adults to medical attention more often. |

For the present study, the "Cognitive Load Hypothesis" (see Table 1a) is of particular interest. According to this hypothesis, the extra effort required for auditory sensory/perceptual processing might be a significant cause of faster cognitive decline in hearing-impaired adults. More specifically, a remarkable auditory processing deterioration for spoken language is one of the issues that occurs most frequently with aging. Moreover, age-related HL entails a perceptual decline in acoustic discrimination of time, intensity, and frequency domains, specifically critical for recognition of words, as well as a dysfunctional central auditory integration, specifically critical for sound localization and recognition of

spoken language in a noisy environment. Indeed, difficulties in following a conversation in noisy listening situations represent a typical manifestation of age-related HL [13].

The growing evidence reflecting HL as a controllable risk factor for dementia has supported the research aiming to unveil the potential benefit of aural rehabilitation in cognitive performance. Conventional auditory rehabilitation offers hearing aids for moderate to severe HL and cochlear implants (CI) for severe to profound HL [8,11,14]. Regardless of age, CIs constitute the most suitable and valid auditory prosthetic solution to restore functional hearing in people with severe to profound sensorineural HL [15,16].

The effect of hearing restoration on cognitive functions in CI users is a relatively new topic. Indeed, until 2015, there was a lack of prospective studies on the assessment of postoperative cognitive functions in elderly CI users [17]. However, in the last seven years, several studies have investigated this topic, e.g., refs. [3,14,16,18–23]. These studies found controversial results: some of them indicated postoperative cognitive improvement, e.g., ref. [3], while some others did not observe any significant performance improvements, e.g., ref. [22]. A review by Claes et al. [24] reported that the majority of the existing studies (five out of six) resulted in a significant postoperative cognitive benefit [18–20,25,26] whilst only one study did not observe any significant performance improvement [21].

Conversely, the efficacy of CIs in speech perception has been widely studied and recognized. It is considered the best practice for a wide range of ages, including the elderly population, although the extent of the benefit is highly variable [27]. Among predictive factors, demographic (older age at implantation), audiological (duration of HL, decline in spectral resolution and sensitivity to temporal cues, the amount of preoperative/postoperative residual hearing), and surgical factors (positioning of the electrode array and the angle of insertion) [27–30] seem to account for 10 to 20% of the variability in CI outcomes [31].

The role of neuropsychological functions in elderly CI outcomes has been emphasized by a growing number of studies and their findings might be crucial for a deeper understanding of the link between HL and cognitive decline, e.g., [1,14,23,27,30,32–34]. At present, the outcomes from these studies do not offer a basis for solid conclusions owing to multiple factors. Among them, two factors that are very important for the goals of the present study need to be discussed in detail here.

The first factor is linked to the construct of "cognition" and "cognitive skills" [35] as well as to the variability in the assessment tasks. Cognition is "information processing" and encompasses several functions of different complexity, ranging from subcortical stimuli processing to basic (attentional control and memory) and higher-order executive functions [35]. Indeed, cognition is a very complex construct, an "umbrella term" including several domains and subdomains. Hence, the results of studies on this topic should be interpreted by considering their conceptual and methodological framework.

In the context of research on CI outcomes, the most studied cognitive function has been working memory (WM): there is broad consensus that verbal WM is a cognitive function significantly involved in speech recognition by normal hearing and hearing-impaired subjects [36–39], and that its capacity declines with aging [27,40]. WM is a dynamic memory system that is able to temporarily store and process various information, necessary for complex cognitive tasks such as comprehension, learning, reading text, problem-solving, and reasoning [41]. Despite the presence of various WM conceptualizations with different theoretical and research proposals, most models recognize WM as a dual mechanism consisting of short-term storage and an information processing component [27]. Audiological researchers investigating the role of WM in speech recognition have used a wide variety of tasks for the assessment of the same construct, ranging from less demanding tasks (e.g., a forward digit span) to more demanding ones (e.g., a backward digit span or reading span task). These tasks differ from each other significantly, as they focus on different aspects of cognitive functioning. As underlined by Moberly et al. [27], a forward digit span test primarily assesses the storage component of WM (*how many elements are you able to recall?*) whereas a backward digit span involves more processing components (*how many elements are you able to recall in inverse order?*) [42]. Hence, the results of WM studies in

hearing-impaired people may partly depend on the theoretical framework and the type of task. A second factor is the difficulty of isolating one cognitive domain from another when studying their effects on individual CI outcomes. Among cognitive domains, WM and its role in speech recognition outcomes has been the most common research topic [27]. Especially in noisy listening contexts, WM compensates for the degraded auditory signals provided by amplification systems and/or CI [34]. However, it should also be considered that WM tasks always involve some level of attention [40,43] and so far, the impact of attention on WM performance might have been ignored. Attention is not a unitary concept [44] but a complex construct of various definitions and conceptualizations [43,44]. Nevertheless, in general terms, attention can be defined as the ability of our perceptual system to select the information of interest to us within our processing capacity, acting as an input filter [45]. In this sense, attention is seen as a mechanism of "priority selection" for the information to be processed, e.g., ref. [44]. From another theoretical perspective, attention is defined as a limited resource of information processing [43]. Theories conceptualizing attention as a resource assume that this resource is responsible for the limited capacity of WM [43] and this association between WM and attention is also referred to as executive attention [46].

General cognition, WM, and attention are conventionally considered predictors of better CI performance [18,27,34]. In clinical and experimental contexts, the difficulty of disentangling the assessment of one cognitive domain from another presents itself at any age but becomes even more evident in elderly people with sensorineural HL. This is mainly due to the impairment of contextual hearing and the aging of the brain, associated with slower cognitive processing and a decline in attentional resources [47]. Thus, the present study aimed to investigate some specific cognitive domains in elderly CI users and to try to disentangle the effects of these cognitive skills on speech perception. Considering HL as a significant risk factor for cognitive decline [5,6,11], a better understanding of audiological aspects in the elderly CI population could enable early/special intervention of cognitive functions, providing a potential benefit for postoperative cognitive and audiological outcomes [3,48].

## 2. Materials and Methods

### 2.1. Participants and Study Design

The present study consisted of elderly CI users, all implanted and regularly followed at the CI center of the Sapienza University of Rome (Policlinico Umberto I). Informed consent was obtained from each subject prior to study enrollment, and all procedures were approved by the local ethics committee of the Sapienza University of Rome (Protocol no: 5982, 22.04.2020).

The inclusion criteria for study enrollment were being ≥60 years of age and having >12 months of CI experience. The data collected included age at implantation, gender, side of implantation, listening mode (unilateral, bilateral, or bimodal), duration of HL, etiology, aided sound-field (SF) audiometry, speech perception assessment in both quiet/noise, and cognitive performance in a memory, attention, and reasoning test. The exclusion criteria regarded as significant a self-reported history of psychiatric conditions and/or diagnosed incident dementia as well as a cognitive and anxiety level outside the normal clinical range. Elderly CI users with any comorbid medical conditions potentially impacting neuropsychological functioning including stroke, ischemic attack, traumatic brain injury, and concussion were excluded from this study.

Thirty postlingually deafened CI users, all native Italian speakers, participated in the present study. All unilaterally and bilaterally implanted (UCI and BCI, respectively) participants had bilateral severe to profound HL. Bimodal users (CI and a contralateral hearing aid) (BIM) showed severe to profound HL in the implanted ear and a down-sloping moderate to severe HL on the hearing aid side.

All 30 participants successfully completed the cognitive assessment battery; hence, their data were all included in the statistical analysis. Participants (16 M and 14 F) had a mean age of 73.4 (range 60 to 87 years; SD = 6.6). Twenty participants (66.7%) had

a basic education (primary and lower secondary); eight (26.7%) had an intermediate upper secondary level, and two (6.7%) had an advanced educational level (bachelor's or equivalent degree).

The mean duration of HL—defined by self-reported first hearing aid use—was 36.7 years (range 6 to 70 years; SD = 16.4) and the mean duration of CI experience was 8.6 years (range 1 to 22 years; SD = 5.54). A total of 15 participants (50%) were unilateral CI users, whilst 5 participants (16.7%) had bilateral CI and 10 participants (33.3%) were bimodal (CI/HA) users. A total of 12 participants had AB® cochlear implants (1 CII HF 1J, 5 HiRes 90K adv, 3 HiRes 90K HF1J, and 3 HiRes Ultra MS receivers; all using Naida 90 BTE Processors) fitted with Optima™ strategy; 14 participants used Med-El devices (1 Pulsar CI-100 and 13 Synchrony receivers; all using Sonnet BTE Processors) fitted with FS4 strategy; 4 received Cochlear® devices (2 CI24RE, 1 CI512, and 1 CI632 receivers; all using Nucleus 7 BTE Processor) fitted with ACE™ strategy. The descriptive data of the participants are shown in Table 2.

**Table 2.** Descriptive data of the participants (n = 30).

| Personal Variables | | Mean (sd) |
|---|---|---|
| Age at test (years) | | 73.4 (6.6) |
| Duration of HL (years) | | 36.7 (16.4) |
| CI experience (years) | | 8.6 (5.54) |
| | | **n (%)** |
| CI listening mode | Unilateral | 15 (50.0) |
| | Bilateral | 5 (16.7) |
| | Bimodal (CI/HA) | 10 (33.3) |
| Gender | Male | 16 (53.3) |
| | Female | 14 (46.7) |
| Status | Married | 23 (76.7) |
| | Unmarried | 2 (6.7) |
| | Widow | 5 (16.7) |
| | Living alone | 4 (20.0) |
| | Living with significant others | 16 (80.0) |
| Educational level | Basic | 20 (66.7) |
| | Intermediate | 8 (26.7) |
| | Advanced | 2(6.7) |

Abbreviations: CI = cochlear implant; HA = hearing aid.

*2.2. Procedure*

The procedure consisted of an audiological evaluation followed by a cognitive assessment.

2.2.1. Audiological Assessment

An audiological assessment was performed with participants' everyday listening mode. Pure tone SF audiometry was measured both for unilateral and bilateral/bimodal listening conditions. The assessment was performed in a sound-proof audiometric chamber using an Aurical audiometer (Otometrics Taastrup, Denmark) connected to a loudspeaker placed at 0° azimuth at 1 m distance from the participant's head.

Speech perception tests consisted of Italian disyllabic words (W) and sentences (S) [49]. Tests were administered in quiet (Wq/Sq) and in noise with a fixed signal-to-noise ratio (SNR) at +10 (W+10/S+10) and +5 dB (W+5/S+5). Both speech and noise signals were presented from a loudspeaker at 0° azimuth at a 1 m distance from the participant's head and the primary signal was fixed at 65 dB HL. Testing for each speech material was preceded by a training list.

2.2.2. Cognitive Assessment

All participants were tested individually in a quiet room by a psychologist experienced in the clinical assessment of hearing-impaired patients. To facilitate speech comprehension

during the assessment procedure, the psychologist did not wear an FFP-2 mask, as indicated by government provision. The cognitive assessment was performed in everyday best-aided listening mode.

For the screening of general cognitive functioning and anxiety symptoms, the following assessment tools were used: the Raven Colored Progressive Matrices (CPMs) [50,51], the Montreal Cognitive Assessment [52,53], and the State-Trait Anxiety Inventory for Adults (STAI-Y) [54]. The CPMs have been designed to evaluate the cognitive level of children (with typical/atypical development) and adults with intellectual disabilities or elderly with cognitive decline. The CPMs' normal values range between 25° and 75° percentile and the risk of dementia is considered at <5° percentile. The verbal (culturally based) cognitive functioning was assessed via the Montreal Cognitive Assessment (MoCA) [52]. Normative values are based on an Italian adult population [53]: the test considered a cut-off value of 15.5 (corrected for age and educational level) to be indicative of a significant cognitive decline. The State-Trait Anxiety Inventory for Adults (STAI-Y) [54] is a well-known and easy-to-apply questionnaire for the detection of anxiety symptoms in adult populations. The range of possible scores for each scale varies from 20 to 80, with a predictive threshold value of severe anxious symptomatology set at 60.

Aside from the screening, a psychometric test battery was implemented to assess short-term memory, verbal WM, attention domains, and state-trait anxiety symptoms. The cognitive assessment battery consisted of the following:

**Forward and backward digit span** (FDS and BDS): these are the most frequently used tests of short-term memory/simple WM [55]. In the present study, we used the version of De Beni and Borella [56]. The test consists of two tasks: a forward digit span (passive short-term memory) and an inverse digit span task (active short-term memory). The examiner verbally presents digits at a rate of one per second. In the *forward form*, the participant is required to repeat the digits verbatim. In the *backward form*, the participant has to repeat the digits in reverse order. The score is corrected for age, gender, and educational level.

**Categorization working memory span task (CWMT)** [56]. This task was used to assess participants' verbal WM. It consists of 20 lists of 5 words (100 words in total) organized in sets of 3 to 6 lists. Each list contains a maximum of 2 animal names. The participant is required to listen to the lists, memorize the final word of each list, and remember the words in the last position with the correct order of presentation at the end of each set. In addition to the memory task, the participant is required to perform a secondary task which is to beat the hand on the table every time the name of an animal appears. The total number of correctly remembered words represents the verbal WM capacity index (maximum score = 20) [56]. The score is corrected for age, gender, and educational level.

**RBANS attention subsection**: The evaluation of attention capacities was carried out through the administration of subtests from the repeatable battery of assessment of neuropsychological status (RBANS) [57]. The RBANS attention domain consists of two subtests (1 verbal digit span and 1 nonverbal symbol/digit association): the combination of the obtained raw score provides an attention domain index score which is corrected for age at the time of the test. Following the recommendations of Patton et al. [58], who applied cut-offs to the RBANS score to increase scale sensitivity, attention performance was analyzed as standard scores and subsequently, performances were divided into two subgroups: low/medium attention performers (LMAP) (≤90) and high attention performers (HAP) (≥91).

*2.3. Statistical Analysis*

Descriptive and inferential statistical analyses were performed. After checking the normality of each data distribution with both Shapiro–Wilk and Kolmogorov–Smirnov tests, a non-parametric analysis was adopted. Kruskall–Wallis and Mann–Whitney U tests (with Bonferroni correction for multiple comparisons, when necessary) were used to compare the effects of listening mode (UCI, BCI, and BIM) and gender on cognitive (FDS, BDS, CWTM, RAS, STAI-Y-1, and STAI-Y-2) and audiological (SF and speech perception) outcomes. The statistical analysis showed no statistically significant effects ($p > 0.05$).

Therefore, the participants were not divided into subgroups with respect to the variables of gender and listening mode. Subsequently, Mann–Whitney U tests were performed to compare participants based on attention performance (LMAP and HAP based on a cut-off of 90) [56]. The effect size was calculated using the Rosenthal formula r¼Z/N (small effect ¼ 0.10–0.30, moderate effect ¼ 0.30–0.50, and large effect >0.50) [59].

For statistical analysis, the percent correct scores for speech perception in quiet were transformed into rationalized arcsine units (RAUs) to avoid the ceiling effects [60].

Finally, a Spearman correlation analysis was performed to evaluate the associations between cognitive variables, and a simple regression analysis investigated the relationships between cognitive and audiological variables; $p$-values $\leq 0.05$ were considered statistically significant. For multiple comparisons (SF thresholds and word/sentence recognition) with Bonferroni correction, $p \leq 0.017$.

## 3. Results

### 3.1. Outcomes of Attention Assessment and Their Comparison with Audiological/Cognitive Data

The median attention values for the LMAP and HAP subgroups were 74 (range 58 to 89) and 100 (range 90 to 126), respectively. The median SF threshold for octave frequencies between 250 to 4000 Hz was 31 dB HL (range 20 to 55 dB HL). The corresponding value was 35 dB HL (range 20 to 50 dB HL) at 250 Hz, 30 dB HL at 500 Hz (range 20 to 40 dB HL), 30 dB HL at 1000 Hz (range 20 to 45 dB HL), 30 dB HL at 2000 Hz (range 20 to 45 dB HL), and 35 dB HL at 4000 Hz (range 20 to 55 dB HL).

A total of 93% of subjects showed a score > 50% for word and sentence recognition in quiet. Word recognition scores from the LMAP and HAP subgroups were: 80% (22 to 100%) versus 90% (range 58 to 100%) in quiet, 40% (range 0 to 55%) versus 60% (range 0 to 90%) at SNR+10, and 0% (range 0 to 20%) versus 28% (range 0 to 88%) at SNR+5. Sentence recognition scores from the LMAP and HAP subgroups were: 90% (range 0 to 100%) versus 90% (range 70 to 100%) in quiet, 40% (range 0 to 70%) versus 60% (range 0 to 100%) at SNR+10, and 0% (range 0 to 10%) versus 20% (0 to 100%) at SNR+5.

For the cognitive variables, the comparison between the LMAP and HAP subgroups showed statistically significant differences for both CWTM and DIGIT performance. Participants with high attentional performance had significantly higher values than low performers for all WM tests (Table 3). Except for CWTM (effect size = 0.42), all statistically significant differences showed a medium/large effect size (0.5–0.64).

### 3.2. Correlations and Regressions

Spearman's correlation analysis was carried out to see which sociodemographic and audiological data were significantly related to cognitive variables (Table 4). Strong correlations were observed between attention and all other cognitive variables ($0.534 \leq$ Rho $\leq 0.742$, $p < 0.01$). Statistically significant correlations were found between attention and both SF ($-0.404 \leq$ Rho $\leq 0.581$, $p < 0.01$) and speech perception in noise outcomes ($0.397 \leq$ Rho $\leq 0.664$). Interestingly, the results showed a statistically significant correlation between attention and educational level (Rho = 0.604) whereas for CWTM significant correlations were observed only with W+10 and W+5 (Rho of 0.497 and 0.519, $p < 0.05$, respectively).

**Table 3.** The table shows the differences between the LMAP and HAP subgroups for dependent cognitive variables, measured with the Mann–Whitney U test and adjusted with Bonferroni correction for multiple comparisons. Significant correlations are shown in bold. For multiple comparisons (word and sentence recognition) with Bonferroni correction, *p*-values were ≤0.017 (*).

| **A.** *Cognitive Variables* | **LMAP** **Rank Sum** | **HAP** **Rank Sum** | **U** | **Z** | ***p*-Value** | **Effect Size** |
|---|---|---|---|---|---|---|
| **CWTM** | 146.000 | 319.000 | 55.000 | 2.322 | 0.019 | 0.425 |
| **FDS** | 119.000 | 346.000 | 28.000 | 3.452 | 0.000 | 0.642 |
| **BDS** | 137.000 | 328.000 | 46.000 | 2.699 | 0.005 | 0.511 |
| **B.** *Audiological Variables* | **LMAP** **Rank Sum** | **HAP** **Rank Sum** | **U** | **Z** | ***p*-Value** | **Effect Size** |
| **W quiet** | 113.50 | 211.50 | 47.50 | 1.632 | 0.106 * | 0.320 |
| **W+10** | 92.00 | 233.00 | 26.00 | 2.391 | 0.005 * | 0.470 |
| **W+5** | 102.50 | 222.50 | 36.50 | 2.798 | 0.026 * | 0.550 |
| **S quiet** | 127.00 | 198.00 | 61.00 | 0.919 | 0.381 * | 0.180 |
| **S +10** | 112.50 | 212.50 | 46.50 | 1.685 | 0.094 * | 0.330 |
| **S +5** | 126.00 | 225.00 | 48.00 | 2.017 | 0.064 * | 0.400 |

Abbreviations: LMAP = low–medium attention performers group (RBANS attention subscale < 90); HAP = high attention performers group (RBANS attention scale ≥90); RBANS = repeatable battery of assessment of neuropsychological status; CWTM = categorization working memory span task; FDS = forward digit span; BDS = backward digit span; W = words, S = sentences; +5, +10 = signal-to-noise ratio.

For the whole group, the simple linear regression analysis with Bonferroni correction showed a statistically significant linear dependence between attention and aided SF at 2000 Hz (R = 0.611, $R^2$ = 0.379, *p* = 0.005) and 4000 Hz (R = 0.753, $R^2$ = 0.567, *p* ≤ 0.0001) as well as a trend toward significant linear dependence with aided SF at 250 Hz (R = 0.489, $R^2$ = 0.235, *p* = 0.035). Attention significantly predicted speech perception in noise for W+10 (SNR+10: R = 0.530, $R^2$ = 0.281, *p* = 0.017) (Figure 1) while there was a trend of significance for W+5 (SNR+5: R = 0.538, $R^2$ = 0.289, *p* = 0.019) and sentences (SNR+5: R = 0.470, $R^2$ = 0.221, *p* = 0.04; SNR+10: R = 0.470, $R^2$ = 0.221, *p* = 0.04) (Figure 2). Simple linear regression based on the RBANS score explained more than 50% of the variance for word recognition in noise and more than 46% of the variance for sentence recognition in noise.

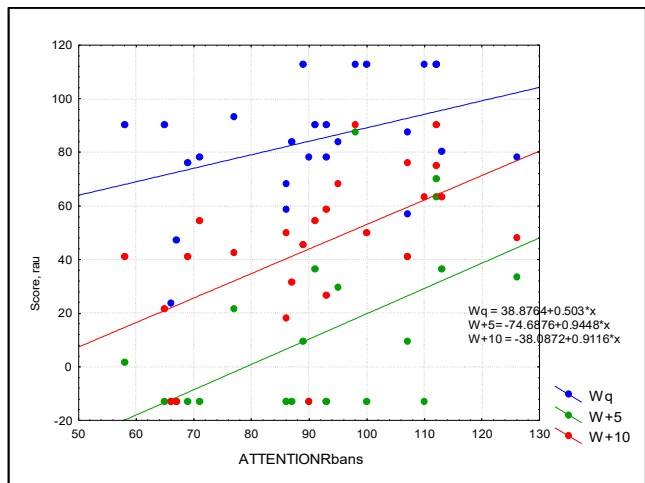

**Figure 1.** Scatterplot of audiological outcomes for words in three conditions: blue = quiet (Wq), green = SNR = +5 (W+5), and red = SNR+10 (W+10) as predicted by attention in the study sample (n = 30). Simple linear regression based on the RBANS score explained more than 50% of the variance for word recognition in noise.

**Table 4.** Relationships between sociodemographic, cognitive, and audiological variables in the overall study group (n = 30).

| VARIABLES | SF 250 Hz | SF 500 Hz | SF 1000 Hz | SF 2000 Hz | SF 4000 Hz | Wq | W+10 | W+5 | Sq | S+10 | S+5 | Age at Test (yrs) | Education (yrs) | HL duration (yrs) | Attention | CWTM | FDS | BDS |
|---|---|---|---|---|---|---|---|---|---|---|---|---|---|---|---|---|---|---|
| **Age at test (yrs)** | **0.43 *** | 0.32 | 0.23 | 0.29 | 0.15 | −−0.25 | **−0.47 *** | **−0.52 **** | **−0.44 *** | −0.15 | −0.27 | −− | **−0.38 *** | −0.27 | −0.34 | −0.30 | −0.07 | 0.002 |
| **Education (yrs)** | **−0.56 **** | **−0.40 *** | **−048 *** | **−0.42 *** | **−0.53 **** | 0.25 | 0.38 | **0.43 *** | 0.37 | 0.1 | 0.25 | −0.37 | −− | −0.09 | **0.60 **** | 0.30 | 0.27 | **0.53 **** |
| **HL duration (yrs)** | 0.00 | −0.34 | −0.15 | −0.19 | −0.16 | 0.08 | 0.32 | 0.15 | 0.30 | 0.45 | 0.23 | −0.27 | −0.10 | −− | −0.12 | −0.00 | −0.27 | −0.05 |
| **Attention** | **−0.58 **** | **−0.40 *** | −0.37 | **−0.45 *** | **−0.47 *** | 0.356 | **0.66 **** | **0.58 **** | 0.32 | 0.34 | **0.40 *** | −0.34 | **0.60 **** | −0.19 | −− | **0.67 **** | **0.74 **** | **0.53 **** |
| **CWTM** | **−0.50 *** | **−0.41 *** | −0.33 | −0.38 | −0.28 | 0.31 | **0.50 *** | **0.52 *** | 0.07 | 0.28 | 0.33 | −0.31 | 0.31 | −0.00 | **0.66 **** | −− | **0.56 **** | 0.32 |
| **FDS** | **−0.40** | −0.29 | −0.32 | −0.32 | −0.27 | 0.06 | **0.40** | 0.38 | −0.12 | 0.25 | 0.27 | −0.07 | 0.27 | −0.27 | 0.74 | **0.56 **** | −− | **0.46 *** |
| *BDS* | −0.31 | −0.36 | **−0.41 *** | −0.39 | **−0.54 **** | 0.07 | 0.27 | 0.25 | −0.06 | 0.27 | 0.22 | 0.00 | 0.53 | −0.05 | **0.53 **** | 0.32 | **0.46 *** | −− |

\* = correlation (Rho) statistically significant at $p < 0.05$; \*\* = correlation (Rho) statistically significant at $p < 0.01$. Significant correlations are in bold. Abbreviations: HL = hearing loss; CWTM = categorization working memory span task; FDS = forward digit span; BDS = backward digit span; SF = sound-field audiometry; W: words, S: sentences, q = quiet condition; +5 = signal-to-noise ratio +5; +10 = signal-to-noise ratio +10.

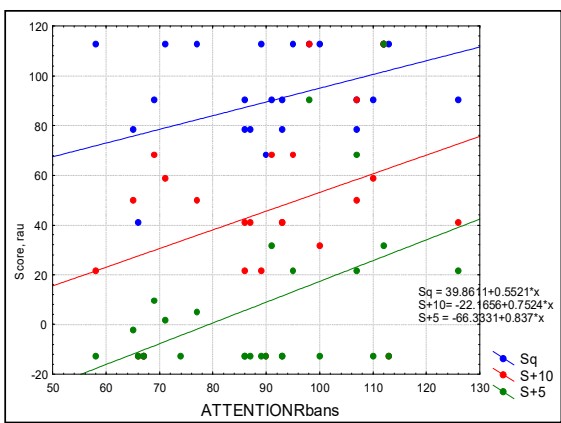

**Figure 2.** Scatterplot of audiological outcomes for sentences in three conditions: blue = quiet (Sq), red = SNR = +10 (S+10), and green = SNR+5 (S+5) as predicted by attention in the study sample (n = 30). Simple linear regression based on the RBANS score explained more than 46% of the variance for sentence recognition in noise.

Following post hoc analysis, WM showed a trend toward a significant prediction of W+10 (R = 0.43, $R^2$ = 0.19; *p* = 0.031) and W+5 (R = 0.47, $R^2$ = 0.224; *p* = 0.019) (Figure 3).

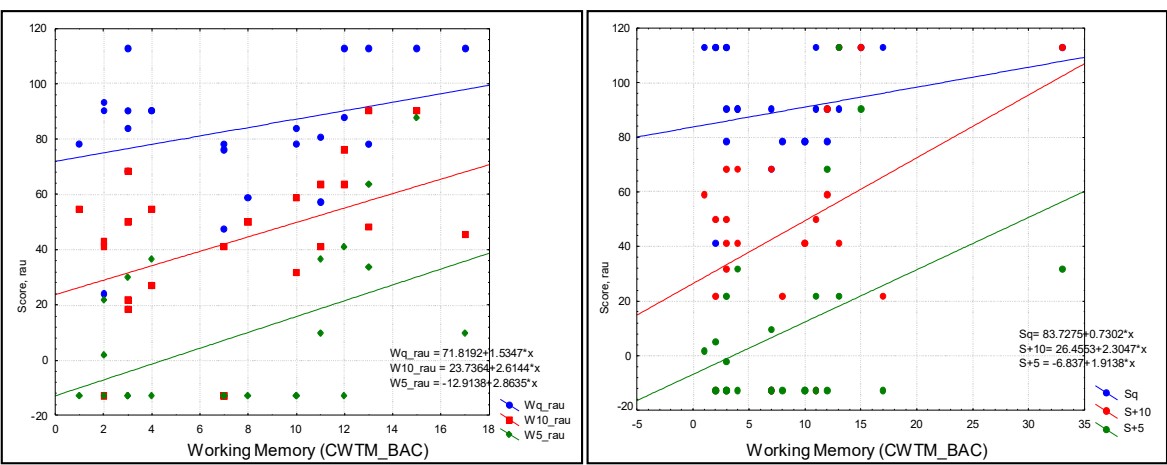

**Figure 3.** Scatterplot of audiological outcomes. Left: words in three conditions (blue = quiet (Wq), red = SNR = +10 (W+10), and green = SNR+5 (W+5)). Right: sentences in three conditions (blue = quiet (Sq), green = SNR = +10 (S+10), red = SNR+5 (S+5)) as predicted by WM in the study sample; scatterplots were produced without the contribution of outliers in the analysis: W+10 (R = 0.43, $R^2$ = 0.19; *p* = 0.031) and W+5 (R = 0.47, $R^2$ = 0.224; *p* = 0.019).

## 4. Discussion

Cognitive domains such as attention, cognition, and WM are believed to contribute significantly to better CI outcomes, especially in postlingually deafened elderly populations, e.g., refs. [14,31–34]. Hence, the present study aimed to gain insight into the effects of these cognitive functions on postoperative speech perception outcomes for an elderly CI population. Not surprisingly, the present findings reflected the significant link between attention, cognition, and WM. Moreover, attention resulted in playing a significant role in outcomes from both tonal and speech audiometry in noise whilst WM was significantly related to speech perception in noise for this sample of elderly CI users. More specifically, the important role of cognitive functions became more apparent for open-set recognition of words in noise than that of everyday sentences, where the increase in speech material's semantic predictability might have provided a remarkable perceptual benefit for reducing

the cognitive load and listening effort as attentional effort, despite the challenges of listening in noise [61].

Overall, the findings highlighted that better cognitive performance may positively contribute to better speech perception outcomes, particularly for complex listening situations where noise is present, and performance relies more on the perception of auditory cues such as open-set words instead of highly predictable everyday sentences. Such results were further supported by the present significant correlations between attention and aided SF audiometry, which was observed for the first time specifically in elderly CI users. In other words, higher attention scores predicted lower (better) CI thresholds or vice versa. Similar to a previous paper by Mancini et al. [62], the present sample showed aided SF thresholds in line with those obtained from typical adult CI users. Likewise, even for a group of CI listeners who achieved target PTAs better than 40 dB HL, the significant effects of aided SF thresholds on a speech recognition test (the STARR test) presenting everyday sentences at low, medium, and high levels in adaptive noise were reported by Dincer D'Alessandro et al. [63]. Moreover, better STARR findings were observed for a group of elderly bimodal users, benefiting from the summation effect to improve overall bilateral audibility when listening both with a CI and a contralateral hearing aid [62]. Indeed, perceptual difficulties for speech at the low level, probably increasing listening effort linked to degraded auditory input requiring greater auditory attention as in the case of shorter duration and less predictable word recognition test in noise here, appeared to be the major factor limiting STARR performance despite the highly predictable everyday sentences of the test [63,64]. Similarly, a significant difference between low and high attention performers for word recognition at SNR +10 was found in the present study group. Despite the increased number of CI users showing floor effects for word recognition at SNR + 5, a trend toward a significant difference was observed for this test condition as well. Hence, it is reasonable to expect a significant difference for a larger population instead of the present LMAP (n = 13) versus HAP (n = 17) subgroup comparisons with a smaller sample size. Indeed, the use of Bonferroni correction in such a small sample might have been too conservative due to the risk of Type 2 statistical error linked to an increased probability of producing false negatives [65]. Such an expectation is also supported by a good correlation index found by bivariate correlational analysis, reflecting statistically significant correlations between attention and speech perception in noise for both words and sentences in the overall group (n = 30). Despite losing significance after the Bonferroni correction, a trend toward significant dependence was still present for the linear regression analysis as well.

Recently, the effects of cognition and attention on speech perception performance have been emerging topics in the field of cochlear implantation. However, the existing literature is limited to a few studies. This fact might partly stem from the limited number of available tools to assess such specific cognitive functions specifically in elderly hearing-impaired populations. There are a few screening tools, such as the RBANS and MoCA, used for the assessment of cognitive domains, including cognition and attention in relation to auditory skills in elderly CI recipients. The RBANS is a well-known neuropsychological tool for clinical diagnosis and follow-up of dementia and mild cognitive impairments. The test offers a total score of cognition, which can also be divided into five cognitive domains, including attention. A newer version of this tool, RBANS-H, has been specifically developed for hearing-impaired people, with the aim of avoiding the bias linked to HL, especially in the elderly population [66]. On the other hand, the MoCA is a rapid screening tool to measure cognition and addresses questions regarding memory recall and executive functions. The tool evaluates components concerning delayed memory, visuospatial abilities, executive functions, language, attention, and orientation [52].

Unlike the present work, previous studies using RBANS-H [3,67] and MoCA [68] did not observe any significant correlations between attention and speech perception in noise. Such outcome differences might be partly owing to differences in study samples and methodology. More specifically, the present work used the original version of RBANS instead of RBANS-H designed specifically for hearing-impaired people. The RBANS

attention task adopted in the present study consists of a nonverbal symbol/digit association and a verbal digit span—this last one was administered in auditory–visual mode with available lipreading cues. Concerning the attention task, both the RBANS-H and RBANS have the same nonverbal symbol/digit association, while the digit span is administered in an auditory–visual modality in the case of RBANS-H. On the other hand, Vasil et al. [68] study used MoCA subscales to investigate correlations with speech perception outcomes. The authors found a significant correlation between the percentage of correct recognition scores for words and the test of delayed recall, for which stimuli were presented in an auditory-alone condition. However, they did not observe any significant correlations with the attention subtests which are administered in auditory-alone conditions as well. The authors speculated that correlations and linear regression models might be interpreted in two ways: cochlear implantation may improve the performance of cognitive functions even for nonauditory stimuli; or, alternatively, individuals with better cognitive skills show better postoperative speech perception performance. In light of the present findings, we believe that differences in correlations between the two auditory tasks might be partly dependent on the cognitive domain, and significant correlations between attention and auditory perception may support the hypothesis that individuals with better attention skills might be those who make better use of auditory cues conveyed by CI and consequently show better postoperative performance.

The present sample was divided into two subgroups of low and high performers (LMAP and HAP) with the aim of gaining insight into the role of attention in other cognitive functions. These subgroups showed statistically significant differences for both the categorization of WM and forward/backward digit span tests. More specifically, participants with high attentional performance had significantly higher values than low performers for all cognitive tests. Moreover, attention performance was highly correlated with WM, indicating a medium/high coefficient. This result appears highly consistent with the existing literature [43,44,46]. There is a strong link between WM and attention [43], and thus, it is possible to state that "by virtue of holding a selected subset of all available representations in memory, WM is by definition a form of attention" [43], p. 14. A main strand of empirically supported theories has conceptualized attention as a limited resource for information storage and processing, suggesting attention to be responsible for the limited capacity of WM [43]. Interestingly, a different strand of theories also defined attention as a selection process [69]. In this sense, WM can also be conceptualized as an "instance of attention" [43], p. 7: whilst selective attention is "attention to perceptual objects", WM is "attention to memory objects". Finally, WM capacity is believed to be closely related to attention ability in order to focus on a target by excluding any distractor from the encoding process [40]. Indeed, several researchers support the hypothesis that WM capacity might be limited by an attentional resource [43]. Unlike younger adults, in the elderly population, a higher reliance on attention skills might be required for encoding a WM task without distraction.

Age-related cognitive decline negatively affects speech recognition, especially in challenging situations such as listening in the presence of competing noise [6,11]. Therefore, in a complex auditory scenario, cognitive resources are spent on perceptual processing to the detriment of other cognitive processes [70,71]. Hence, it is conceivable that in the HAP subgroup—because of higher attentional resources—the cognitive load needed for storing and processing information might be lower than in the LMAP subgroup, significantly influencing the performance for both simple and complex WM tasks.

As mentioned above, according to the present findings, attention seemed to play a significant role in speech perception, especially for more complex listening tasks: the HAP subgroup showed significantly better outcomes in the perception of words presented in noise (W+10). Similarly, regression analysis showed that attention significantly predicted recognition of words presented at SNR+10. Moreover, for the linear regression analysis of words presented at SNR+5 and sentence recognition in noise, the presence of a trend toward significance even after Bonferroni correction in such a small sample was a promising finding. Indeed, it is a well-known fact that "top-down" cognitive processes are key

elements of speech perception, linking incoming acoustic signals to phonological and lexical representations in long-term memory [27,72]. Nevertheless, cognitive mechanisms involved in speech processing follow different paths depending on whether the incoming sounds are clear (as it happens for normal hearing people and/or in quiet listening situations) or degraded (as it happens for people with sensorineural HL even in the best-aided listening condition, especially in noisy contexts). In fact, current CI technology provides useful acoustic cues in the time, intensity, and frequency domains and a good representation of envelope cues that are required for speech understanding in quiet, whereas temporal fine structure cues, known to be crucial for complex listening such as speech in noise and music, are mostly removed from CI speech processing [13]. As a matter of fact, speech processing strategy is an extremely important aspect of CI technology. Currently, there are various signal processing techniques that differ between CI systems (for more information on this topic, see for example Choi and Lee [73]). Nevertheless, electrical stimulation induces a pattern of auditory nerve activity poorer than acoustic stimulation [74], in particular for coding of low-frequency signals such as the fundamental frequency of voice [75].

According to the ease of language understanding model (ELU) [39], when acoustic signals are clear, incoming sounds are encoded by an implicit effortless speech processing channel. On the contrary, when incoming signals are degraded, perceptual ambiguity increases, and recognition of acoustic information gets harder. In that case, the incoming sounds are encoded by a compensatory, slower, and more effortful explicit channel, that strongly relies on WM. Here, high attention resources are needed to overcome the mismatch between the incoming signal and the encoded one [31,76]. Hence, it is conceivable that better CI performers in terms of speech perception make more robust use of the explicit channel "which requires good working memory and a high level of attention to match the degraded signal with the storage in the long-term memory" [31], p. 549.

Whilst the link between WM and CI outcomes has been widely studied [16,31,34,42,77], the role of attention in CI outcomes has received less attention. Very recently, auditory selective attention has been shown to significantly affect linguistic outcomes in pediatric CI users [78]. Auditory attention largely depends on listeners' ability to enhance the representation of a target auditory source. This requires analyzing the acoustic scene and segregating the target sounds, showing attentional focus on the target, suppressing interfering elements, and simultaneously maintaining cognitive flexibility to switch attention toward new auditory targets required by the context [79]. It is a complex process at any age in the hearing population but becomes even more effortful in CI users, because of the distorted transmission of auditory objects, especially in a noisy environment [78]. It is conceivable that in our sample of elderly CI recipients, higher attentional resources may lead to diminishing cognitive resources required to store and process auditory information (WM), which is especially important for understanding complex speech. In this light, higher attentional resources may allow using attentive skills both for auditory processing and memory representations. Conversely, low attentional resources in the LMAP subgroup might be a significant cause of lower scores for word recognition in noise (both at SNR +5 and +10 dB), where an increased cognitive load is needed to fill the perceptive gaps left by inaudible parts in the acoustic streams [80]. This observation is in line with a study by Volter et al. [31] showing that differences in cognitive functioning and linguistic skills may explain poor speech recognition scores in adult CI performers. Although significant differences between poor and good CI performers in terms of speech perception could be detected in various cognitive subdomains, Volter et al. [31] found that the most prominent difference was observed for the attentional task. Attention had the strongest power to discriminate between poor and good performers in elderly CI users, and it was significantly improved after one year of CI use. Similar to Volter et al. [31], in the present study the correlation analysis seems to reflect a significant role of attention in CI outcomes both for tonal and speech audiometry in noise whereas, for verbal WM (CWTM), the only significant correlation was observed with words at SNR+10 and +5. Furthermore, the present findings from regression analysis showing the presence of a trend toward significance even after

Bonferroni correction supported such arguments. These findings, taken together, seem to support the ELU model [39]. Reducing the extent of mismatch between incoming speech stimuli and their phonological representation in long-term memory, CI provides a reduction in attentional effort. In turn, increased availability of attention might be crucial for WM involvement in speech understanding, especially in noisy environments [23]. On the other hand, it is conceivable that CI users with higher attentional resources may achieve better speech understanding as a result of the significant link between attention and WM [23,43].

Finally, a significant impact of schooling was observed for attention, while no significant associations were found between educational level and WM tasks. These results are inconsistent with Volter et al. [23], who found a significant impact of educational level on a WM task, but no significant impact on attention tasks. Nevertheless, the differences in tasks and CI experience between the study groups may explain such differences. As a matter of fact, in the present study, we used a WM task (CWMT) with a score corrected for age and educational level, whereas the attention task score from RBANS is corrected only for age. Indeed, Claes et al. [67], in their cross-sectional study using the RBANS-H, found a significant positive association between the RBANS-H score and educational level in a sample of experienced adult CI users. It is widely recognized that educational attainment is a key component of successful cognitive aging [81,82]. From a socio-economic point of view, a higher level of education leads to increasing involvement in cognitive-demanding occupations and participation in social/leisure activities that are cognitively more engaging and stimulating.

Interestingly, educational level is strictly linked to the construct of the cognitive reserve [83], a latent construct that has recently been added to the list of modifiable risk factors for cognitive decline/dementia and is made up of several factors with three dimensions (education, work, and leisure activities) [2]. The cognitive reserve is a concept based on brain plasticity and refers to the ability of the brain to cope with damage and aging by using pre-existing cognitive processes or compensatory brain networks [84]. Although the cognitive reserve is mostly indexed by education, it is a concept highly based on brain plasticity; hence, it is made of potentially controllable indicators such as occupational attainment and stimulating activities, e.g., reading, writing, playing music, social engagement, and cognitive exercises [16]. In this light, the construct of the cognitive reserve and the association that we found between attention and educational level might be promising indicators for cognitive interventions in elderly CI users who are potentially able to exploit their residual brain plasticity [85].

## 5. Limitations

A limitation of the present study stems from its methodology, making it impossible to state a causal relationship among the study variables. Moreover, the absence of preoperative cognitive evaluation does not allow us to investigate the cognitive evolution of the present long-term CI users and the effects of the amount of auditory benefit on their cognitive performance.

Another limitation might be a bias linked to the use of RBANS for assessing attention skills in this group of elderly CI users. RBANS-H might have been a better alternative as it is specifically designed for hearing-impaired people to minimize the effects of bottom-up auditory processing on attentional outcomes. However, it is a matter of fact that there is no adaptation of the RBANS-H in the Italian language. On the other hand, auditory verbal tasks may have a greater relevance to the recognition of degraded speech [27,42,78,86]. Indeed, the present sample consisted of experienced adult CI users, all postlingually deafened. Most of them had a word/sentence recognition score in quiet equal to or greater than 80%, and they were allowed to make use of lipreading cues during testing. None of the participants had a history of auditory deprivation in childhood, and their knowledge of the native language appeared to be comparable to that of a hearing adult of the same age. Hence, auditory verbal instructions presented in a silent environment by a clinician experienced in working with CI users might not be a significant bias. However, without comparisons of RBANS and RBANS-H outcomes in such a population, we cannot rule

out a bias linked to bottom-up processing, potentially leading to an underestimation of cognitive functioning [31].

## 6. Conclusions

The present study observed significant associations between attention and auditory outcomes for both aided SF thresholds and speech perception in noise. Moreover, participants with high attentional performance showed significantly higher scores than low attentional performers for all WM tests. The enhancement of WM and attention skills may positively contribute to speech perception performance in elderly CI users. Whilst WM plays a significant role in the storage and processing of auditory verbal stimuli, more robust attention may lead to better performance for speech perception in noise where a high level of auditory selective attention is needed. In view of the present results, the effectiveness of an early intervention based on auditory cognitive training should be investigated in order to improve cognitive and audiological performance in elderly CI users.

**Author Contributions:** Conceptualization, I.G.; methodology, I.G. and M.N.; formal analysis, B.M.S.I., P.M. and A.C.; investigation, I.G., G.P. and A.P.; resources, A.G.; writing—original draft preparation, I.G. and B.M.S.I.; writing—review and editing, P.M. and H.D.D. All authors have read and agreed to the published version of the manuscript.

**Funding:** This research received no external funding.

**Institutional Review Board Statement:** The study was conducted in accordance with the Declaration of Helsinki and approved by the Institutional Review Board of Policlinico Umberto I (Protocol no: 5982, 22.04.2020).

**Informed Consent Statement:** Informed consent was obtained from all subjects involved in the study.

**Data Availability Statement:** The data are available upon request.

**Acknowledgments:** We would like to thank the participants for their valuable contributions to the study.

**Conflicts of Interest:** The authors declare that they have no conflict of interest.

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
