# Peer review of "Neuropsychological Functions and Audiological Findings in Elderly Cochlear Implant Users: The Role of Attention in Postoperative Performance"

_audiolres, doi:10.3390/audiolres13020022_

Round 1

Reviewer 1 Report

Dear authors, Thank you for sharing your work. Please find some suggestions below:
1. Maybe I overlooked this, but could you please specify in which condition (best aided, CI alone), the subjects were examined during the cognitive tests? (2.2.2. cognitive assessments)

2. How was the duration of HL defined, was it from the start of the first HA, was it self-reported? (section 3. results "The mean duration of HL was 36.7 years (range 6 to 70 years; SD=16.4)")

3. In your discussion you state " Cognitive domains such as attention, cognition, and WM are considered as significant predictors of better CI outcomes, ...". Please bear in mind the cognitive tests were performed after CI implantation. Stated as such, it might appear to the reader that these factors were taken into account prior to implantation (because of the wording "predictors"). This hasn't been explored in your current research - please clarify. 

4.. Maybe I overlooked this, but I couldn't find the raw scoring of the SF data per frequency. It would be relevant to provide the outcomes on the SF test before linking these data points with other variables such as cognitive metrics (as per your sentence" In other 44 words, higher attention scores predicted lower (better) CI thresholds." in the discussion). Could poor fitting have led to poorer (higher) SF thresholds and thus poorer attention? It would be interesting to explore the link between fitting (mapping) and SF thresholds. Are there any remarkable findings from the fitting relevant (e.g. very narrow dynamic ranges, deactivation of a number of electrodes)... I would be curious to understand. Thanks for providing more information on this aspect. 

5. I'm curious to understand why the RBANS-H hasn't been implemented instead of the RBANS, as the latter requires auditory processing and thus can hamper the results in a hearing-impaired population. Later on, I read: "Performance on modality-general measures of WM (i.e., auditory-visual) might not be strongly tied to speech recognition, while modality-specific (i.e., auditory alone) WM tasks may more specifically assess a listener’s ability to store and manipulate auditory input.". Thanks you for clarifying. However, again this stresses the importance of measuring the bottom-up processes as well, which hasn't been part of your research. Please include this in the limitations. 

6. In your discussion you state "we believe that differences of correlations between the two auditory tasks might be partly dependent on the cognitive domain, and that significant correlations between attention and auditory perception support the hypothesis that individuals with better attention skills might be those who make better use of auditory cues conveyed by CI and consequently show greater postoperative performance improvement". However, I see no data showing this improvement in your paper, as the pre-op speech perception scores are not included. hence the wording "improvement" might be misleading. It might very well be that a subject has "only" a 5% improvement (from 60 to 65% pre-post-op), vs another subject having a 20% improvement (from eg. 30 to 50%), however the latter subject in doing more poorly. Please consider rephrasing. 

7. Have you considered splitting up your groups into good and poor performers based on their speech perception outcomes, in order to explore the role of neurocognitive factors, instead of dividing them into two groups based on the attention metric? ("Present sample was divided into two subgroups of low and high performers (LMAP 96 and HAP) with the aim to get insight into the role of attention in other cognitive functions"). I'm concerned about how the attention test was performed (relying on auditory input). 

8. As you state "As a matter of fact, speech processing strategy is an extremely important aspect of CI technology", I wonder whether you have considered adding this information for your subjects (which sound processor, which electrode, which speech coding strategy, which maxima, etc...)

9. I might have overlooked this but could it be that the high performers are those who are bilaterally or bimodally aided users? As stated in your discussion ("Nevertheless, cognitive mechanisms involved in speech processing follow different paths depending on whether the incoming sounds are clear (as it happens for normal hearing people and/or in quiet listening situations) or degraded (as it happens for people with sensorineural HL even in the best-aided listening condition, especially in noisy contexts)", I would like to understand the potential differences in the bottom-up decoding for these groups better (see previous remark on possible fitting differences as well in this respect). 

10. In your discussion "better CI performers" line 154, please specify whether this is based on speech outcomes or attention (as you have defined "better" based on attention outcomes and not speech outcomes). Same holds for" Although significant differences between poor and good CI performers could be detected in various cognitive subdomains, the Authors found that the most prominent difference was observed for the attentional task"

Author Response

Dear Editors and Reviewers

Two tables with questions and point-by-point answers are attached. The changes have been marked in red in the main text.

Rewier #1:

COLUMN A

COLUMN B

1. Maybe I overlooked this, but could you please specify in which condition (best aided, CI alone), the subjects were examined during the cognitive tests? (2.2.2. cognitive assessments)

Thanks for the remark. A sentence concerning how patients listen during the cognitive assessment has been added to page 9, line 223

How was the duration of HL defined, was it from the start of the first HA, was it self-reported? (Section 3. results "The mean duration of HL was 36.7 years (range 6 to 70 years; SD=16.4)")

Duration of HL was defined by first self-reported HA use. pag 7. Line 186

3. In your discussion you state " Cognitive domains such as attention, cognition, and WM are considered as significant predictors of better CI outcomes, ...". Please bear in mind the cognitive tests were performed after CI implantation.

Stated as such, it might appear to the reader that these factors were taken into account prior to implantation (because of the wording "predictors"). This hasn't been explored in your current research - please clarify.

Thanks for the observation. We have rephrased the sentence in Discussion section (pag. 17, line 28-30).

4. Maybe I overlooked this, but I couldn't find the raw scoring of the SF data per frequency. It would be relevant to provide the outcomes on the SF test before linking these data points with other variables such as cognitive metrics (as per your sentence" In other 44 words, higher attention scores predicted lower (better) CI thresholds." in the discussion). Could poor fitting have led to poorer (higher) SF thresholds and thus poorer attention? It would be interesting to explore the link between fitting (mapping) and SF thresholds. Are there any remarkable findings from the fitting relevant (e.g. very narrow dynamic ranges, deactivation of a number of electrodes). I would be curious to understand. Thanks for providing more information on this aspect.

All subjects belonging to the study group were evaluated post device revision and in their best fit conditions.

We have already partly answered and commented on this aspect in a previous publication of the senior researcher, which was focused on elderly bimodal CI users. (ttps://doi.org/10.1080/14992027.2020.1843080).

In summary, the majority of elderly bimodal CI users have a higher FF than a younger population, although they tend to benefit from loudness summation when in bilateral or bimodal listening modes. The mean SF of the elderly evaluated in the above work was 36 dB HL, in the range 125-8000 Hz, slightly worse than reported in the literature in younger patients (Potts et al. 2009). This result, as commented in the cited publication, is better than reported by Aimoni et al. (2016) and Benatti et al. (2013). These studies showed a mean PTA between 125 and 8000 Hz ranging from 36 to 44 dB HL in elderly IC carriers.

A possible explanation for higher CI thresholds in the elderly population could be lower electrical loudness tolerance, in part dependent on lower auditory processing for frequency and loudness discrimination than in younger participants (He, Dubno, & Mills 1998; Moore 2014 ).

On the other hand, the lower thresholds found in the present study may be the volume balance accuracy that all subjects recruited for this study underwent.

In any case:

1. The average SF threshold per octave has been added to the results. Page 11 line 289.

2. The mean SF value for the present study is similar to what has already been published. This comment was added to the discussion on page 17.

3. Finally, for this paper, I have not correlated the audiometric results with the electrical thresholds in order not to get off topic, but I agree with you that this could be an interesting further topic for an audiological article on elderly CI users .

5. I'm curious to understand why the RBANS-H hasn't been implemented instead of the RBANS, as the latter requires auditory processing and thus can hamper the results in a hearing-impaired population. Later on, I read: "Performance on modality-general measures of WM (i.e., auditory-visual) might not be strongly tied to speech recognition, while modality-specific (i.e., auditory alone) WM tasks may more specifically assess a listener’s ability to store and manipulate auditory input.".

Thanks you for clarifying.

However, again this stresses the importance of measuring the bottom-up processes as well, which hasn't been part of your research. Please include this in the limitations.

The RBANS Subscale of Attention (adopted in the present study) is made up two tasks (a Visual Digit Substitution Symbol test) – and a Auditory Digit Span Task). The scale offers a Attention Index Score as a combination of the two raw scores. The only difference between RBANS and RBANS-H Attention Subscale is the presentation of the Digit Span Task, presented in a double visual-auditory modality in the RBANS-H form.

Our choice, to use the only auditory version of digit span, as well as a cognitive test battery prominently auditory based, was driven by the following consideration:

in existing literature, the use of RBANS version adapted for hearing impaired persons concerns prospective longitudinal study with pre and post implantation cognitive assessment.  Of course, a neuropsychological evaluation in a severely HI person without CI needs a visual- written adaptation (such as power point presentation). In the present study, we assessed only experienced CI users with an excellent understanding of speech in quiet context. The evaluation protocol we performed was administered by a clinician with long experience in neuropsychological testing in CI population, without FFP2 and in a very quiet room. Moreover, as we wrote in limitation section, most of them had a word/sentence recognition score in quiet equal to or greater than 80%, and they were allowed to make use of lipreading cues during testing. In our opinion, when an experienced CI users’ sample is tested, auditory-verbal instructions presented in a silent environment by a clinician experienced in working with CI users might not be a significant bias.

These suggestions have been rephrased in Limitations.

6. In your discussion you state "we believe that differences of correlations between the two auditory tasks might be partly dependent on the cognitive domain, and that significant correlations between attention and auditory perception support the hypothesis that individuals with better attention skills might be those who make better use of auditory cues conveyed by CI and consequently show greater postoperative performance improvement". However, I see no data showing this improvement in your paper, as the pre-op speech perception scores are not included. hence the wording "improvement" might be misleading. It might very well be that a subject has "only" a 5% improvement (from 60 to 65% pre-post-op), vs another subject having a 20% improvement (from eg. 30 to 50%), however the latter subject in doing more poorly. Please consider rephrasing.

Thanks for the observation, you are right the sentence is misleading and has been rephrased (page 18, line 83-87)

7. Have you considered splitting up your groups into good and poor performers based on their speech perception outcomes, in order to explore the role of neurocognitive factors, instead of dividing them into two groups based on the attention metric? ("Present sample was divided into two subgroups of low and high performers (LMAP 96 and HAP) with the aim to get insight into the role of attention in other cognitive functions"). I'm concerned about how the attention test was performed (relying on auditory input).

We thank the reviewer for the important comment.  the aim of the study was precisely to discover and investigate the link between "cognitive variables" on speech perception outcomes.

Our choice to divide the experimental sample according to the level of attention resulted from an initial exploratory cluster analysis, where the significant variable in dividing our sample into two groups was Attention (p<0.001).  On the contrary, only 7% of the subjects had a words and sentence score in quiet < of 50%, and it was not possible to use speech perception to divide the group into two subgroups. Further, the analysis suggested by Patton et al, (Line 242), was to divide the sample into low and high performers at the attention test in other to improve analysis

8. As you state "As a matter of fact, speech processing strategy is an extremely important aspect of CI technology", I wonder whether you have considered adding this information for your subjects (which sound processor, which electrode, which speech coding strategy, which maxima, etc...)

Yes we did consider to also process data by this more technical audiological aspect, but it would have been definitely off topic, besides being based on insufficient number of data. More detailed information about CI characteristics have been added to Participants section.

9. I might have overlooked this but could it be that the high performers are those who are bilaterally or bimodally aided users? As stated in your discussion ("Nevertheless, cognitive mechanisms involved in speech processing follow different paths depending on whether the incoming sounds are clear (as it happens for normal hearing people and/or in quiet listening situations) or degraded (as it happens for people with sensorineural HL even in the best-aided listening condition, especially in noisy contexts)", I would like to understand the potential differences in the bottom-up decoding for these groups better (see previous remark on possible fitting differences as well in this respect).

We thank the reviewer for the remark. Before proceeding to the statistical comparison by dividing the experimental sample into low and high performers in attention, we conducted the comparisons according to gender and according to acoustic listening mode (unilateral, bilateral, bimodal) (Lines 259-261). As no statistically significant differences emerged, we proceeded, also supported by the results of the cluster analysis (see the reply to comment #7) to the comparison according to the cognitive variable attention. We do not have enough data to shed potential light on listening modes and bottom-up decoding in this group.

In order to make the procedure clearer, we modified the text by inserting the following sentence "Therefore the participants were not divided into groups with respect to the gender and listening variables" (pag 10, Lines 271-272)

10. In your discussion " " line 154, please specify whether this is based on speech outcomes or attention (as you have defined "better" based on attention outcomes and not speech outcomes). Same holds for" Although significant differences between poor and good CI performers could be detected in various cognitive subdomains, the Authors found that the most prominent difference was observed for the attentional task".

Thank you for the observation: in both cases we referred to speech recognition outcomes, and this information was implemented in both sentences.  

Reviewer 2 Report

This study intended to address an interesting quesiton, i.e., the relations between HL and cognitive decline in elderly CI listenersand underlying mechanisms. The topic is timely and much remains to be investigated. However, I do have concerns related to the introduction, hypotheses presented in the current version, statistical analyses, and potential false positives in the results. I hope the comments below can help clearing the confusion when revising the paper. 

Major comments

1.     According to the layout of the introduction, I was expecting the paper to focus on relations between speech perception and working memory, and general cognitive functions. Towards the end, the goal suddenly became to identify the audiological marker?

If the study also aimed to examine the relations between audibility at each frequency and speech perception and/or cognitive ability in elder CI users, it needs to better motivate this point.

This is important as pure-tone thresholds at each frequency were included in the correlational analyses for speech perception and cognitive measures in section 3.3.

2.     I love the table of hypotheses in the literature about cognitive decline and HL in the introduction, but I had a hard time connecting it with the hypotheses in the current study. Some of these were interpreted in the discussion, I wish it can be moved up to the front to support the hypotheses in the current study.

3.     Why is anxiety of interest? I guess a clearer definition of general cognitive ability will be helpful. Otherwise, it looks quite exploratory. Further, when lots of assessments were included in correlational analyses without multiple comparison corrections, false positives in results could be concerning.

4.     Section 3.2. What motivated the division of subjects into LMAP and HAP groups and the comparisons of the differences in WM and speech perception between the two groups?

5.     Section 3.3. It seemed to me that multiple comparison corrections were not conducted for all the correlational analyses. Some of the results may not be significant any more after corrections.

6.     Results in Figure 3 (left), i.e., correlations between WM and word recognition seemed to be driven by the one subject on the top right. Are the result still significant after taking out the ‘outlier’?

Minor comments

7.     Elderly was defined as >65 years of age in the introduction; but in the method (during recruitment), elderly was then defined as 60 years of age. Please be consistent.

8.     Participant information in section 2.1 and that in section 3.1 can be merged.

9.     Section 2.2.1. How many words and sentences were tested? How were word and sentence perception scored?

10.  Section 2.3. ‘The statistical analysis did not show any statistically significant effects (p>0.05), and therefore, the participants altogether were considered as a single sample.’ Why not include gender and listening mode as covariates and run partial correlations? The non-significant result could be due to the small sample sizes in each subgroup?

Author Response

Rewier #2:

The changes have been marked in red in the main text.

COLUMN A

COLUMN B

1.       According to the layout of the introduction, I was expecting the paper to focus on relations between speech perception and working memory, and general cognitive functions. Towards the end, the goal suddenly became to identify the audiological marker?

If the study also aimed to examine the relations between audibility at each frequency and speech perception and/or cognitive ability in elder CI users, it needs to better motivate this point. This is important as pure-tone thresholds at each frequency were included in the correlational analyses for speech perception and cognitive measures in section 3.3.

We thank the reviewer for the observation and here we will try to add clarity to the point.

The paragraph of the introduction to which you referred was meant to be a brief overview of the most studied neuropsychological variables in relation to outcomes in the elderly with cochlear implant and in particular on what are the main methodological, theoretical and practical limits for an in-depth understanding of the link between the variables described above in relation to cognitive decline in the elderly.

This last paragraph has therefore been slightly modified for a better understanding of the relationship between working memory and attention and the choice to investigate these cognitive abilities. The phrase "finding potential markers of cognitive decline related to audiometric findings" was a reference to the relationship between cognitive variables and speech understanding outcomes.

However, we have eliminated this additional sentence as potentially confusing (line 156, page 6).

As for the second observation, this was also a request from the first reviewer. Information and comments have been inserted in the appropriate sections. (Mean SF threshold per octave has been added to the results. Page 11 line 289. Discussion comment has been added on page 17, threshold per octave is already included in the correlation analysis.

2. I love the table of hypotheses in the literature about cognitive decline and HL in the introduction, but I had a hard time connecting it with the hypotheses in the current study. Some of these were interpreted in the discussion, I wish it can be moved up to the front to support the hypotheses in the current study.

Thanks for the comment. We have now added a paragraph in the introduction to better explain the tables. In the introduction, it was intended to provide the reader with a synthetic theoretical framework of the possible causal and non-causal mechanisms that potentially connect the two conditions. It was part of an introductory description of the state of the art.

However, if you deem it more appropriate, we can eliminate the summary table and provide some references related to the topic.

3. Why is anxiety of interest? I guess a clearer definition of general cognitive ability will be helpful. Otherwise, it looks quite exploratory. Further, when lots of assessments were included in correlational analyses without multiple comparison corrections, false positives in results could be concerning.

Thanks for the interesting question.

A definition of cognition was incorporated into the manuscript on p. 5. Line 111-116.

As for the question "Why is anxiety interesting?" several reasons led us to pay attention to the symptomatology of anxiety in the sample of the present study. I will try to summarize them in the following two points:

- A higher level of anxiety has been found to be a risk factor for age-related neurocognitive disorders, greater cognitive decline, and an increased risk of dementia; (see for example: Ilardi et al., 2021 for extended references).

- Anxiety is known to negatively impact cognitive performance, so we decided to evaluate the presence of anxiety symptoms to check their impact on cognitive performance. However, no significant relationships were found between Anxiety and WM and Attention scores in the present study; we found support for our choice to use a hearing-based prevalent battery to assess WM/attentional resources, consistent with other studies of NH people that did not find a significant impact of anxiety on WM verbal tasks (Walkenhorst and Crowe 2009).

Finally, as no significant relationship was found, we have now eliminated the anxiety as variable in the correlational analysis  

We have reformulated a section in paragraph 2.2.Procedure (line 224-235) as all the subjects were within the normal range (line 173).

4. Section 3.2. What motivated the division of subjects into LMAP and HAP groups and the comparisons of the differences in WM and speech perception between the two groups?

The division into low and high scores was applied with reference to Patton et al., 2006 (Line 260, p.10) which used cut-offs for the RBANS score to increase the sensitivity of the scale. Consequently, following Patton's observations, the study sample was divided into artists with low and high attention.

5. Section 3.3. It seemed to me that multiple comparison corrections were not conducted for all the correlational analyses. Some of the results may not be significant any more after corrections.

Thank you for this observation. We have considered now Bonferroni correction for multiple comparisons and updated our Table and Results accordingly. Group comparisons for W10 remained statistically significant.

6. Results in Figure 3 (left), i.e., correlations between WM and word recognition seemed to be driven by the one subject on the top right. Are the result still significant after taking out the ‘outlier’?

We thank the reviewer for his acute observation.

We have re-conducted the statistical analysis by removing the outlier you observed. Accordingly, this has been specified in the new scatterplot legend. Although the value of R and R2 decreased, they remained significant in both W+10 (R=0.43, R2=0.19; p=0.031) and W+5 (R=0.47, R2=0.224; p=0.019). We report below, for clarity and completeness, the regression scatterplot illustrating the Working Memory variable as a significant predictor of word perception in noise at +5 and +10 (Figure 3 left in the manuscript) without the outlier, and we thank you again for allowing us to further investigate our data.

7. Elderly was defined as >65 years in introduction; but un the method, elderly was the defined as 60 years of age. Please be consistent.

Thanks for the observation. We corrected where necessary.

8. Participant information in section 2.1 and that in section 3.1 can be merged

The paragraphs have been merged.

9. section 2.2.1 How many words and sentences were tested? How were word and sentence perception score?

The Italian words and sentence tests are made of lists with 20 items. Score was calculated in % for correct word and sentence count respectively. For statistical analysis was corrected by the appropriate formula into RAU score.

10. 10.  Section 2.3. ‘The statistical analysis did not show any statistically significant effects (p>0.05), and therefore, the participants altogether were considered as a single sample.’ Why not include gender and listening mode as covariates and run partial correlations? The non-significant result could be due to the small sample sizes in each subgroup

We did the partial correlation analysis with non-significant variables, but the correlation was not changed.

Round 2

Reviewer 2 Report

Dear authors, thank you so much for addressing the comments in the revised version. I really appreciate the clarity in the introduction. 

However, the multiple comparison correction issue has not been carefully addressed in the revised version. Specifically, please note whether p-values in Tables 3 and 4 were adjusted or unadjusted. Further, p-values on page 14 and page 15 were still not adjusted? I found it conflicting that there were no differences in most of the speech perception tasks between groups with low and high attention performance (Table 3), but correlational results suggest that attention and speech ability is correlated. if p-values were adjusted, then this won't be a problem? 

Author Response

Dear Reviewer, 

please see the attachment in the box below.

Round 3

Reviewer 2 Report

I'd like the thank the authors for their responses. Acoording their adjusted p thretholds (p<0.017), then on page 14,  attention would be correlated with speech perception in noise (neither for words nor for sentences). 

On page 15, WM did not predict word perception in noise, either. 

I think the authors needed to be clearer about unadjusted and adjusted p-values throughout the result section. I think it is okay to report 'non-significant' results after multiple comparison corrections, then discuss the trends and limitations in the discussion. 

Author Response

Thanks for your further indications. Please see the attachment for our point-by-point response.
